**Title**

Confined vs. extended Dirac surface states in topological crystalline insulator nanowires

**Authors**

Roni Majlin Skiff[1], Fernando de Juan[2,3], Raquel Queiroz[4], Subramanian Mathimalar[4,]
Haim Beidenkopf[4*], Roni Ilan[1*]

[1]Raymond and Beverly Sackler School of Physics and Astronomy, Tel-Aviv University, Tel-Aviv 69978, Israel.

[2]Donostia International Physics Center, P. Manuel de Lardizabal 4, 20018 Donostia-San Sebastian, Spain.

[3]IKERBASQUE, Basque Foundation for Science, Maria Diaz de Haro 3, 48013 Bilbao, Spain

[4]Weizmann Institute of Science, Israel

[*]Corresponding authors

**Abstract**

Confining two dimensional Dirac fermions on the surface of topological insulators has remained an outstanding conceptual challenge. Here we show that Dirac fermion confinement is achievable in topological crystalline insulators (TCI), which host multiple surface Dirac cones depending on the surface termination and the symmetries it preserves. This confinement is most dramatically reflected in the flux dependence of these Dirac states in the nanowire geometry, where different facets connect to form a closed surface. Using SnTe as a case study, we show how wires with all four facets of the $< 100 >$ type display pronounced and unique Aharonov-Bohn oscillations, while nanowires with the four facets of the $< 110 >$ type such oscillations are absent due to strong confinement of the Dirac states to each facet separately. Our results place TCI nanowires as a versatile platform for confining and manipulating Dirac surface states.

**Introduction**

The interplay between symmetry, geometry, and topology in quantum materials allows for novel states to be synthesized, and offers countless possibilities for engineering and manipulating quantum effects in mesoscopic systems and devices. These materials provide access to fundamental effects difficult to observe and study in other settings, and offer new functionalities by demonstrating unique transport, mechanical and optical properties. In parallel, tuning mechanisms such as controlled breaking of space or time reversal symmetry, make it possible to gain a high level of control over such effects. This potential has not been fully realized to date. In particular, the surfaces of three-dimensional topological insulators host Dirac fermion states responsible for most of their unique functionality. These typically are extended on the two-dimensional surface. Tuning the properties of the surface has been an outstanding problem in the field, where different hetero-structures have been predicted to function as interferometers or switches based on their low energy surface excitations. Specifically, confining Dirac fermions has proven to be challenging.

In this work, we explore topological crystalline insulator (TCI) materials in a mesoscopic geometry and show that depending on how TCIs grow and cleave, their surfaces can naturally host confined two dimensional anomalous Dirac fermions. Until now such a configuration of surface states has required complex hetero-structures, coupling strong topological insulator surfaces to

magnetism (1) or a dimensional reduction, and hence not been realized in other topological phases. Here we show that these states emerge naturally in TCIs due to controlled symmetry breaking.

TCIs (2-4) have their topological properties arising from the presence of certain crystal symmetries. This raises the question of how these can be used for innovative tuning mechanism, and what novel states will emerge when breaking and restoring such symmetries, combined with time reversal. Recent studies have shown that strain can induce higher-order topology in TCIs (5), and be used to engineer various electronic states (6,7). In parallel, topological insulator (TI) nanowires have been explored in recent years as promising candidates for applications such as quantum switching and quantum computing (8-16). For a strong TI nanowire, the band structure and transport properties of the surface states is tunable by flux: it features a gap which can be closed by applying a magnetic field parallel to the wire, resulting in a perfectly transmitted mode around half integer values of flux. Tuning the chemical potential or flux controls the number of modes at the fermi level and places TI nanowires as a promising platform for switching and controlling of current channels. When placed in proximity to an s-wave superconductor and with the application of flux, topological superconductivity is expected to emerge (11,12,17).

We demonstrate the effect of geometry on confinement in TCIs by examining TCI nanowires with different lattice terminations and cross sections. We focus on the canonical example of SnTe and we show that different choices for lattice termination radically affect the response of the wires to magnetic flux. In particular, we show that for certain geometries, surface states are extended across the wire's surfaces and exhibit novel Aharonov-Bohm (AB) oscillations, while other geometries lead to confinement of the Dirac fermions on the wire's facets. This practically freezes the response to flux and eliminates the AB oscillations. The existence of confined vs. extended states depends on the arrangement of the surface Dirac cones, and on the particular spatial symmetries that are broken at the hinges connecting two surfaces. Such breaking of symmetries introduces mixing of the Dirac points which can introduce strong gaps at the corners and lead to confinement. While the effect is demonstrated for SnTe, we argue that it applies in a much more general setting, and can be explained using general symmetry considerations.

Our findings are supported by a combination of analytical and numerical calculations. The Results section is organized as follows. In Section I we discuss a low-energy model for a TCI in a in a cylindrical geometry, and predict the AB response in these wires from the simplest considerations. In Section II we analyze the band structure of the SnTe nanowires for different sizes and geometries, via tight-binding calculations. We contrast between two types of lattice terminations, and show that one of them shows AB oscillations, while the other does not. In Section III we discuss the results from symmetry considerations relevant for SnTe.

**Results**

**I.       General considerations: multiple Dirac points in a compact geometry**

The fate of AB oscillations and confinement vs. deconfinement of the Dirac surface states of TCIs can be understood by considering the basic building blocks of their low energy theory and how they are accommodated in a compact geometry. To demonstrate how compactification of multiple Dirac points is different from that of a single Dirac point positioned at a time reversal invariant momentum (TRIM) as in strong TI, we begin with a simplified low-energy model of a TCI in a cylindrical wire geometry under flux. We first account for the behavior of the finite size quasi-one-dimensional resolved energy bands of a single Dirac cone located around a generic point in the Brillouin zone (BZ) as a function of flux. The only remaining crystal

symmetry at this stage, unlike the case in ref. (18), is that we still assume that the surface respects the bulk symmetries despite being of cylindrical shape (an assumption we will relax later on).

We therefore begin with a low-energy model of the form $H = \hbar v_x (k_x - k_{x0})\sigma_x + \hbar v_z (k_z - k_{z0})\sigma_z$. Here, $v_x, v_z$ are the Fermi velocities in $x$ and $z$ directions, $k_x, k_z$ are the surface momenta, $(k_{x0}, k_{z0})$ is the location of the Dirac cone, and $\sigma$ are a set of Pauli matrices. Assuming that the wire is perfectly cylindrical with a radius $R$, we perform a coordinate transformation (17,19) and take the wire to be along $z$ axis, with $x$ coordinate going around the wire. The magnetic flux enters the Hamiltonian as an Aharonov-Bohm phase, a shift to the angular momentum $l$, where $\Phi = \frac{\phi}{\phi_0}$ is the total flux quanta through the wire's cross section. The spectrum of the transformed Hamiltonian is then given by

$$E = \pm \hbar \left[ v_x^2 \left( \frac{l+\Phi}{R} - k_{x0} \right)^2 + v_z^2 (k_z - k_{z0})^2 \right]^{1/2} \qquad (1)$$

where the quantum number $l$ takes values of the form $l = n + \frac{1}{2}$ with $n$ an integer, due to the anti-periodic boundary conditions. Writing the momentum shift $k_{x0} = \frac{\alpha}{a}$, with $a$ the lattice constant and $\alpha$ a number, we note that a gapless point will appear in the one-dimensional spectrum, located at $k_z = k_{z0}$ (momentum along the wire) at flux values given by

$$\Phi = \alpha \frac{R}{a} - l \qquad (2)$$

In strong TI wires, the Dirac cone is constrained to sit at a TRIM and therefore $\alpha$ always vanishes. In contrast, in TCI wires $\alpha$ is a material parameter determined by the crystal structure and the perturbations that shift the position of the cones from the high-symmetry points (20). The flux values that generate a gap closing therefore depend on the position of the cones in the surface BZ, as well as the wires' cross section. Hence, this low energy model predicts a different pattern of Aharonov Bohm oscillations as a function of flux.

The model containing a single Dirac point on a cylinder oversimplifies the description of a TCI nanowire in several ways. First, a TCI wire is usually not cylindrical. In order to host gapless surface states, it must have a geometry in which the wire's surfaces respect some non-trivial crystalline symmetry of the material (2,18). The specific bulk symmetries that are preserved or broken on the facets of the wire determine the exact composition of the surface states on each facet, namely the number, chirality, and location of the surface Dirac cones. Second, a non-cylindrical wire has hinges and corners, which break the full rotational symmetry of the cylindrical wire, but may also break the surface symmetries. This results in the angular momentum $l$ no longer being a good quantum number, but becomes rather conserved up to an integer number reflecting the discrete rotational symmetry should it exist. Additional symmetry breaking will introduce further mixing to the surface bands.

The breaking of rotational symmetry exists also in strong TI wires where it does not crucially affect AB oscillations. This is not the case for a TCI wire. The crystalline symmetries impose additional Dirac cones in each surface's BZ, with positions related to the cone at $(k_{x0}, k_{z0})$ by symmetry operations. A simplistic description of the low-energy model of the surface of a TCI wire as a superposition of these cones, results in a modified AB oscillation pattern that is a superposition of the one described above, with more gap closings. However, as we show next, due to the breaking of symmetries close to the corners of the wire, additional terms can be added to the

surface Hamiltonian in the wire geometry. These terms can, in some cases, confine the Dirac surface states per surface, thereby effectively disconnecting them from one another. This confinement unlocks the possibility of studying the Dirac fermions on a bounded two-dimensional flat space.

In the following section we study specifically the flux response of SnTe wires, and we show that the interplay between the number and location of the surface Dirac cones in different wire configurations lead to the behavior described above.

## II.     Flux response in SnTe wires

To demonstrate the differences in flux response arising from confined vs. deconfined surface states in TCI, we now turn to consider two nanowire configurations of SnTe. SnTe was reported and confirmed to be a TCI (21,3,4), as well as a higher-order topological insulator (HOTI) (5), due to the non-trivial mirror Chern number $n_M = -2$ of the crystal $\{110\}$ mirror planes. As a result, protected gapless states would exist on surfaces and hinges that are invariant under one or more of the bulk $\{110\}$ planes. Such lattice terminations will host an even number of Dirac cones, that can be located off the TRIM (20,22). The diversity of the surface states of SnTe allows us to consider different configurations and surface state composition and demonstrate how those affect the physics discussed above.

We model a SnTe crystal which is infinitely long along the cubic crystal's $z$-axis ($[001]$ direction), and finite in the two other directions, using a tight-binding model, as described in the Materials and Methods section. Since the system is periodic along $z$, $k_z$ remains a good quantum number. Next, we demonstrated how the spectrum of this quasi-one-dimensional system depends on the type of terminations, and the overall symmetries of the wire.

## A. SnTe nanowire in the $(100)$ configuration: Aharonov Bohm oscillations

The first configuration considered is of an infinite wire in the $z$ direction, with two surfaces in the $[100]$ direction and two in the $[010]$ direction. It will be referred to as the $(100)$ wire, and is depicted in Fig.1(A). This configuration was recently shown to be experimentally accessible (23,24). In this case, each of the facets in the $[100]$ direction and in the $[010]$ direction respect two bulk mirror symmetries: $(011), (01\bar{1})$ and $(101), (10\bar{1})$, respectively. Therefore, each facet will host, in a slab geometry, four Dirac cones in the surface BZ, located at $(k_{x/y}, k_z) = (k_0, k_0), (-k_0, k_0), (k_0, -k_0), (-k_0, -k_0)$, see Fig.1(F). These mirror symmetries are preserved on the surfaces of a large wire, but would be broken close to the corners. Another important feature of the square $(100)$ wire is the existence of hinge states: the corners of the wire are in the $[110]$ and , $[1\bar{1}0]$ directions, and are invariant under $(1\bar{1}0)$ and $(110)$ mirror symmetries, respectively, see Fig.1(G). Therefore, they are expected to host four pairs of helical hinge modes (5).

The crystalline symmetries of the $(100)$ wire presented above raise two questions: what is the fate of the two-dimensional surface states at the corners of the wire, and how do they interplay with the hinge modes? In strong TI wires, the single Dirac cones are essentially unaffected by the corners of nanowires and nanoribbons due to the topological protection imposed by time reversal symmetry. This is not the case for TCI, which is different both in terms of locally lifting the protection as well as in terms of the number of Dirac cones per surface that may mix at the corners.

We diagonalize the Hamiltonian in Eq.(5) for a square $(100)$ wire with 46 atoms in each finite direction. A sketch of the cross section of the wire is shown in Fig.1(C). In the spectrum

obtained at zero flux, shown in Fig.1(H), all bands are doubly degenerate. We identify the surface Dirac cones, which are gapped due to the finite geometry and, presumably, also by the breaking of mirror symmetry of each surface near the hinges. We also identify four pairs of hinge modes.

When adding the magnetic flux, the double degeneracy of the bands is removed. The spectrum, which is symmetric around the point $k_z = \pi$ at zero flux, remains symmetric for every flux value, see Fig.S2 in the supplementary material. This symmetry exists because the two surface Dirac cones located at $(k_0, k_0), (-k_0, k_0)$ are projected onto the same $k_z$ point along the axis of the wire. While according to the low energy model presented in Section I each Dirac cone will respond differently to the flux, there are two additional Dirac points with interchangeable roles located on the wire's axis on the other side of $k_z = \pi$ such that the total spectrum remains symmetric.

When increasing the flux, we observe a gap closing at a flux value which is neither zero nor half of a flux quantum, but around $\phi = 0.3$ and $\phi = 0.7$ (Fig.2). In addition, when calculating the spectrum for square wires with increasing size, a similar gap closing and re-opening is observed (Fig.2). This is the behavior expected from the effective model, in which the gap closing in the one-dimensional spectrum depends on the dimension of the wire (in the simple cylindrical case- the radius), the location of the Dirac cone on the surface and the number of flux quanta. Interestingly, the gap closing occurs between surface states and hinge state, and not at the gapped surface Dirac cones. We note, however, that in the vicinity of the surface gap, the wavefunctions belonging to bands identified as hinge modes are in fact spread over the entire two-dimensional surface of the wire and become essentially indistinguishable from those of the higher surface bands. This is unlike the situations of a strained wire where the 2D surface bands are gapped on the entire facet of the wire due to symmetry breaking rendering the wire to a HOTI. In a strained wire, the hinge modes are further isolated from the other surface bands and their one-dimensional nature is manifested also close to the gap. In our case, only when moving away from the small surface gap, these hinge modes regain their one-dimensional character, see Fig.3. This observation makes the gap closing between the surface bands and the hinge modes slightly less mysterious and compatible with the intuition gained by the effective modes described in Section I where hinge modes are absent. We note also that close to half of a flux quantum threaded through the wire, some, but not all of the bands restore a double degeneracy. The reason this degeneracy is restored close to, and not exactly at half flux quantum, is due to the finite penetration depth of the surface states into the bulk (11,17,25).

Our analysis of the $(100)$ wire is compatible with predictions for AB as a sum of those arising from projections of several Dirac points with a finite size gap tunable by flux, as described in Section I. The physics is richer due to an interplay of the surface and hinge modes at the corners of the wire, clearly demonstrates that the surface is 2D in character, and sensitive to flux via modification of the boundary conditions. Next, we turn to explore a wire with a different lattice termination, where the behavior markedly different.

**B. SnTe nanowire in the $(110)$ configuration: surface state confinement**

The next configuration we examine is that of an infinite wire along the $z$ direction with two facets in the [110] direction and two in the [1$\bar{1}$0] direction, as depicted in Fig.1(B), which will be referred to as the $(110)$ wire. These facets are crystallographically equivalent and invariant under $(1\bar{1}0)$ and $(110)$ mirror symmetries, respectively. In a slab geometry, each such surface is expected

to host two Dirac cones in the surface BZ, located in $(k_1, k_z) = (0, k_0), (0, -k_0)$, see Fig.1(G), where $k_1 = \frac{k_x + k_y}{\sqrt{2}}$. The hinges of this wire do not respect any additional non-trivial symmetry of the bulk.

We diagonalize the Hamiltonian in Eq.(5) for the (110) wire configuration, for a square wire with a size of 28x28 atoms in the outermost layer. A sketch of the cross section of the (110) wire is presented in Fig.1(D). We emphasize that the wire is periodic in $z$ direction, and due to the rock-salt structure of the lattice, the cross sections for consecutive layers will be of alternating atoms. For example, the cross section of the next atomic layer of the wire in Fig.1(D) will have Te atoms in the outermost layer. This implies that these wire terminations are not atom-type dependent.

The spectrum at zero flux is shown in Fig.1(I). All bands are doubly degenerate. As with the (100) wire, the spectrum remains symmetric around $k_z = \pi$, see Fig.2S in the supplementary material. Tuning the flux away from zero flux and up to one flux quantum, we note a clear distinction in the behavior of the low vs. high energy surface bands: while the low energy surface bands experience a weak response to the flux and appear to be hardly moving, the upper bands experience the conventional pair switching (see Fig.4) similar to the flux response observed both in the (100) wire as well as strong TI wires. Zooming in on the low energy bands (insets of Fig.4 first row), we observe that the bands are arranged in groups of four nearly degenerate bands, which are pairwise degenerate at zero flux. At half flux quantum, two of the four bands restore this degeneracy, while the other two are left non-degenerate. Tuning away from these flux values, the bands do experience pair switching, but this switching is confined within the four-band subspace. This suggests a different picture from the one for the (100) wire.

The conjecture, which will be supported by additional numerical calculations below, is that the in contrast with the (100) wire in which the surface states at all energies extend over the two-dimensional surface of the wire and slightly perturbed by its corners, the surface of the (110) wire is characterized by having each of the wire's facets hosting a confined Dirac semi-metal. Namely, the (110) wire's surfaces represent four copies of the same surface that are slightly mixed at the corners of the wires. Indeed, when examining the (110) spectrum, the twofold degenerate bands come in "pairs" of bands that are very close in energy. It can be observed that these partners, or a pair of pairs, result from fourfold degenerate bands that were mixed when projected on top of each other onto the one-dimensional axis of the wire- four "copies" of the same spectrum. The addition of flux through the wire removes the degeneracy associated with time-reversal, and as the flux approaches half flux quantum, pair switching occurs, and two of the four bands restore the twofold degeneracy. This behavior can be easily captured by a simple ``toy model'' of four sites, each site representing a single surface, with a weak coupling between them, as detailed in the supplementary material.

The observation of weak coupling between states that are localized on each surface can be supported first by examining the spectra of (110) wires with various sizes, shown in Fig.5. First, looking at spectra of square wires with different size, we observe that the overall features remain similar, while the bands move closer in energy for increasing wire's cross section. This is in contrast with the (100) wire where the change in cross-section can introduce gap closings, as discussed extensively in Section II.A.

Second, we consider rectangular wires. If the conjecture of effectively decoupled surfaces is correct, then we expect the bands to split into two sets, corresponding to opposite surfaces with gaps determined by the width of each surface. This is a result of the $C_4$ symmetry being reduced to $C_2$. Note that in a rectangular wire each face still holds the same properties in terms of cone arrangement and symmetry conservation, the only difference is in the width of the two faces. We fix the width of the wire in one direction and examine the bands structure as we increase the width of the perpendicular direction. The effect of this change is shown in Fig.5. Focusing on the low energy bands, half of the bands indeed remain unchanged, while half of the bands move closer in energy (similar to what occurs when increasing size of square wire). If two bands overlap, they mix and open a small gap. This behavior further supports the claim that the low bands in the spectrum of the (110) wire contains four spectra, resulting from the four facets, which are weakly coupled.

To conclude this section, the spectrum of the (110) wire and its flux response is drastically different from that of the (100) wire. While in the (100) wire the flux induces a mixing and gap closing between surface and hinge states, a richer variation of the behavior of a strong TI wire, the (110) wire shows completely different behavior and a much weaker flux response. The surface states on the (110) are "localized" on each of the faces, and are connected via weak coupling, similar to a system of four sites described by a tight binding model, in which hopping terms between the sites slightly lifts the degeneracy of the energies on each site.

## III.   Understanding confinement from symmetry considerations

To understand why certain geometries allow for surface states confinement while some do not, we now consider the differences between the two types of wires in terms of their symmetries. As we argue, the symmetries that break at the corners of the wires conspire with the number of cones projected onto a particular momentum along the translationally invariant direction (the axis of the wire) to allow for particular gaps at the corner to open, that in turn result in effective confinement only in one case but not the other. To solidify the simplified arguments of Section I we refer in particular to the two geometries of SnTe discussed in this text, though a generalization in straight forward.

We start from the (110) wire. As previously stated, each of the wire's facets, when considered in the infinite size limit, hosts two Dirac cones, related by mirror symmetries, time reversal symmetry, and a $C_2$ rotation around the surface's normal. The surface theory of a surface in the $[1\bar{1}0]$ crystallographic direction, in the infinite surface limit, is captured by the low energy Hamiltonian (20)

$$H(k_1, k_2) = (v_1 k_1 \sigma_2 - v_2 k_2 \sigma_1) + m\tau_1 + \delta \sigma_1 \tau_2 \qquad (4)$$

hosting two Dirac points sitting at $\vec{k} = (0, \pm k_2^0)$. This is the low energy model of the surface spectrum depicted in figure Fig.1(G), with $k_1 = \frac{k_x + k_y}{\sqrt{2}}$ and $k_2 = k_z$. The Pauli matrices $\vec{\sigma}, \vec{\tau}$ represent the spin and flavor degrees of freedom, respectively. Given these definitions, the surface respects two mirror symmetries, represented by the operators $M_1 = -i\sigma_1$ and $M_2 = -i\sigma_2 \tau_1$, along with the operators representing twofold rotation around the surface's normal, $C_2 = -i\sigma_3 \tau_1$, and time reversal $T = i\sigma_2 K$ as usual, with $K$ standing for complex conjugation. Note the two Dirac cones have the same chirality, and are mapped into each other by $M_2$ and $C_2$.

Compactifying the wire in the $k_1$ direction is an action that breaks $M_1$ and $C_2$ locally near the corners. Each Dirac cone is broken into a set of energy bands, that are projected onto the one-dimensional axis of the wire and disperse as a function of $k_2$. The mirror symmetry $M_2$ that relates the two cones to one another is still preserved in the wire geometry. While the two terms $m\tau_1, \delta\sigma_1\tau_2$ are the only masses allowed to be added to the Hamiltonian Eq.(4) when all symmetries of the infinite surface are intact, the breaking of $M_1$ and $C_2$ allows for an additional mass term to be added, of the form $\gamma\sigma_3\tau_2$, with $\gamma$ a parameter determined by the details of the lattice and the corner. This term, when added to the low energy Hamiltonian Eq.(4), creates a gap for both Dirac cones. To respect mirror symmetry, the masses must be of opposite sign for the two cones.

The additional gap in each Dirac cone introduced by breaking one of the mirror symmetries, is added to the gaps generated due to the finite-size effects, as described in the effective model in Section I, however, it does not depend on angular momentum or magnetic flux, and therefore is not affected or closed with the application of magnetic field along the wire. Such a gap, if large enough, confines the Dirac fermion states onto each surface. Note that these considerations are only compatible with the low energy theory, and therefore are not expected to hold for energies beyond the linear regime. Indeed, from the numerical calculations it is obvious that the confinement is manifested in the low energy bands (see Fig.4). The overall symmetries of the wire geometry, such as the bulk mirror symmetries combined with any rotation symmetry around the wire's axis, need to respected, allowing for a particular mass configuration to appear in terms of their spatial distribution. For details on the particular mass configurations see supplementary material.

Now consider compactifying the low energy model in Eq.(4) in the orthogonal direction, namely in the direction of $k_2$ coordinate, thereby breaking $M_2$. Since $M_2$ is now broken, but $M_1$ is preserved, only a term of the form $\gamma\sigma_0\tau_3$ can be added to the Hamiltonian. This term does not open a gap in the Dirac cones, but only shifts their location along $k_2$. In this case, the gaps in the spectra of the cones due to finite-size effects are the dominant ones, and the usual flux response is expected with gap closures in flux values that are compatible with the momentum shift of the cones.

The (100) wire is more complex, but encapsulates the two behaviors described above. When considered in the infinite size limit, it hosts four Dirac cones, located at $(\pm k^0, \pm k^0)$, related by four mirror symmetries and a fourfold rotation symmetry around the surface's normal. In consistence with the axis of the two-dimensional BZ in Fig.1(F), compactifying the wire in the $x$ direction is an action that breaks the mirror symmetries $M_x, M_{xz}, M_{x\bar{z}}$ and $C_4$ near the corners of the wire. The breaking of these symmetries may introduce additional terms that would mix and gap the Dirac cones when projected to the one-dimensional BZ of the wire and disperse around $k_z = k^0$ and around $k_z = -k^0$. The added mass terms may introduce shifts of the Dirac cones, that will result in two minima split around $k_z = \pm k^0$, (which can indeed be observed in Fig.2), as well as open magnetic-like gaps. The magnitudes of these masses will be determined by the microscopic details of the material and the specific geometry of the nanowire. However, if the magnitude of the terms that shifts the Dirac cones is significant, they can eliminate the effect of the magnetic-like masses, and leave the surface theory sensitive to the flux threaded through the wire even at low energies.

**Discussion**

Our work suggests that TCI wires are a rich platform that supports confined or extended Dirac surface states, depending on the lattice termination and the associated symmetry breaking. As a result, TCI wires display different flux responses, including modified AB oscillations which are beyond a simple generalization of the flux response in strong TI, or absence of oscillations. To determine which of the scenarios applies, one must take into consideration the number of Dirac cones appearing at each facet, and how they mix due to the breaking of the spatial symmetry at the corners. This is translated into the mixing of cones projected onto one another in the quasi-one-dimensional band structure of the wires, and manifested in the ability of the finite size resolved bands to respond to flux. Most of these features can be reproduced from effective low energy models, but the magnitude of the mixing terms at the corners strongly depends on the microscopic details of the wire.

The mixing of Dirac cones has also been demonstrated to result in one-dimensional states at step edges on surfaces of the TCI $Pb_{1-x}Sn_xSe$ (26,27). A recently published work (28) also discussed SnTe nanowires. However, it considered a single wire geometry, and did not treat the behavior of surface states in the confined wire geometry, or the expected Aharonov-Bohm oscillations when applying a magnetic field parallel to the wire. Another recent publication (29) predicted Aharonov-Bohm oscillations in a different system, that of a chiral HOTI with one-dimensional hinge channels.

In this work we have considered for simplicity wires with four identical facets, with the same lattice termination and Dirac cone arrangement. This allowed us to single out cleanly the two extreme scenarios discussed above. It is possible to consider cases where the facets are different, that may introduce more novel features intermediate to the two extremes studied here.

Confinement of Dirac fermions on the surfaces of three-dimensional strong TI can in principle be achieved by introducing magnetism which, as mentioned earlier on, has proven to be hard. Such confinement has been achieved in Graphene (30-32), by means of spatially limiting to small geometries or by creating electrostatic gating. The latter requires coupling to external materials such as superconductors, gates, or substrates. The alternative presented here based on TCIs and relying solely on the geometry of the wire represents a breakthrough and has the potential to affect their use as well as have implications for the fate of induced superconducting states in such wires. In addition, in contrast with Graphene, while the surface of a TCI also hosts multiple Dirac cones, they are non-degenerate and can be anomalous.

**Methods**

For the purpose of the simulation we use a tight binding model for SnTe (21):

$$H = m \sum_j (-1)^j \sum_{r,\alpha} c_{j\alpha r}^\dagger \cdot c_{j\alpha r}$$

$$+ \sum_{j,j'} t_{i,i'} \sum_{\mathbf{r},\mathbf{r}',\alpha} c_{j\alpha r}^\dagger \cdot \hat{\mathbf{d}}_{r,r'} \hat{\mathbf{d}}_{r,r'} \cdot c_{j'\alpha r'} + h.c. \qquad (5)$$

$$+ \sum_j i\lambda_j \sum_{r,\alpha,\beta} c_{j\alpha r}^\dagger \times c_{j\alpha r} \cdot s_{\alpha\beta}$$

with the following parameters (5,33): $m = 1.65, t_{11} = -t_{22} = 0.5, t_{12} = 0.9, \lambda_1 = \lambda_2 = 0.7$. The model describes a rock-salt lattice with staggered on-site potential on the two atoms, spin orbit coupling, and hopping between three p-orbitals in each atom represented by three components of the $c^\dagger, c$ operators. The operator $c_{j\alpha r}^\dagger$ creates an electron on lattice site $r$, sublattice $j = 1,2$ (for Sn, Te respectively) and spin $\alpha =\uparrow,\downarrow$. The bulk spectrum is therefore described by a 12x12 $k$-

space Hamiltonian. The vector $\widehat{\boldsymbol{d}}_{\boldsymbol{r},\boldsymbol{r}'}$ is a unit vector pointing in the direction of hopping between atoms in $\boldsymbol{r}$ and $\boldsymbol{r}'$. The matrices $\widehat{\boldsymbol{d}}_{\boldsymbol{r},\boldsymbol{r}'}\widehat{\boldsymbol{d}}_{\boldsymbol{r},\boldsymbol{r}'}$ describe $\sigma$-bond hopping between nearest $(t_{12}, t_{21})$ and next nearest $(t_{11}, t_{22})$ neighbors, neglecting $\pi$-bond hopping.

The magnetic field is added to the model as phases to the hopping terms: $t \rightarrow te^{\varphi_{ij}}$. Using the Peierls substitution, a phase of the form

$$\varphi_{ij} = \frac{e}{\hbar}\int_{r_j}^{r_i} \boldsymbol{A} \cdot d\boldsymbol{l}$$

will be added to the term describing hopping from site $\boldsymbol{r}_i$ to site $\boldsymbol{r}_j$, with $\boldsymbol{A}$ the vector potential and $d\boldsymbol{l}$ an element in the direction of the hopping process.

**Data availability**
All data generated and analysed during this study are included in this published article (and its supplementary information files).

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

## Acknowledgments

The authors wish to thank Nurit Avraham, Alon Beck, and Maia G. Vergniory for multiple useful discussions.

RMS and RI are supported by The Israel Science Foundation grant no. 1790/18.

RI is also supported by the BSF grant no. 2018226.

FJ acknowledges funding from the Spanish MCI/AEI/FEDER through grant PGC2018-101988-B-C2.

HB acknowledges support from the European Research Council (ERC-StG no. 678702, "TOPO-NW")

## Author contributions

RI, FJ and HB devised the idea for this project. RMS performed the analytical calculations and the tight-binding numerical calculations. RMS, RI, FJ, RQ and HB worked out symmetry considerations and contributed to understanding the numerical results. SM contributed to discussions. RMS and RI wrote the manuscript. FJ, RQ and HB contributed to reviewing and editing the manuscript.

## Competing interests

Authors declare that they have no competing interests.

**Figures and Tables**

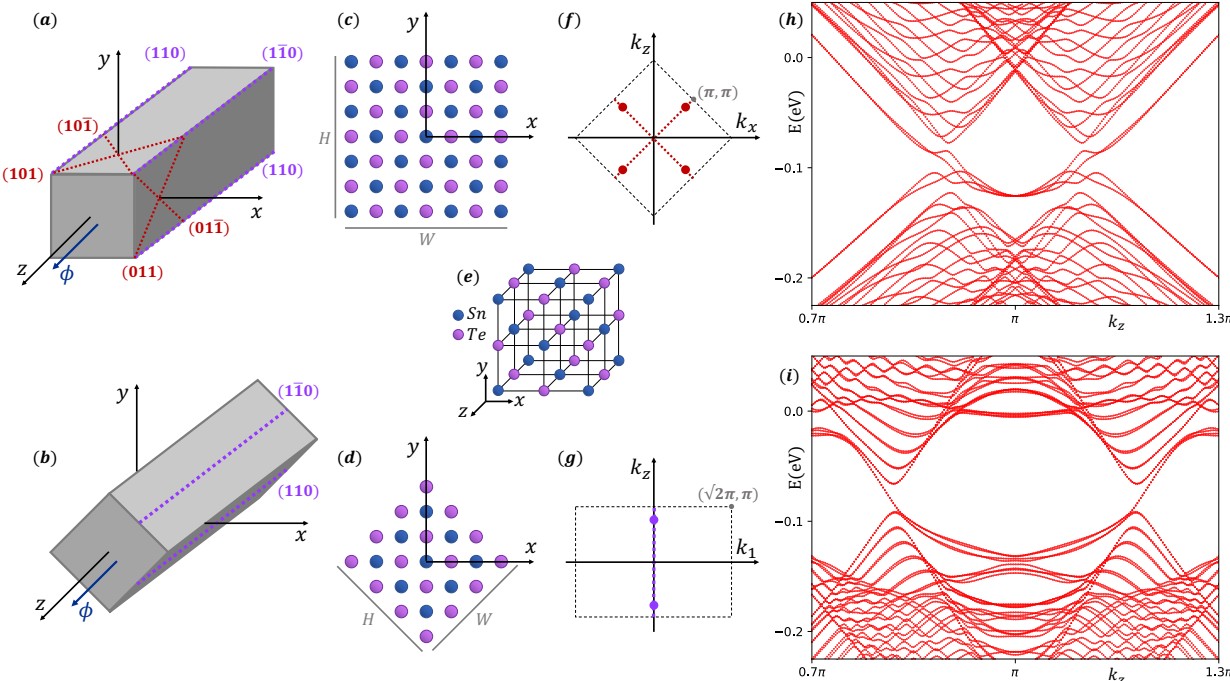

**Fig. 1. Wire configurations.** Suggested configurations for a SnTe nanowire: **(A)** the (100) wire, and **(B)** the (110) wire. The non-trivial mirror symmetries of the surfaces and hinges in each wire are indicated and marked by dotted lines. A flux $\phi$ is inserted along the wire's $z$-axis. **(C)** Cross sections of the (100) wire, shown with 7 atoms in height and 7 in width. Blue and purple dots correspond to Sn and Te atoms. **(D)** Cross sections of the (110) wire, shown with 4 atoms in height and 4 in width. We note that for the purpose of this calculation, the wire's cross section was chosen such the outermost layers are always the one atom longer, so that the hinges are a sharp step without a missing atom. For example, in order to make the 4x4 (110) wire depicted in Fig.1(D) into a 5x4 wire, a layer of three Te atoms and four Sn atoms should be added to one of the facets. In this method, the intersection between the facets remains the same for all wire sizes. **(E)** The bulk's crystal structure. Surface BZs and the existing Dirac cones of **(F)** a plane in the [010] direction, with four Dirac cones on the lines invariant under (101) and (10$\bar{1}$) mirror symmetries, and **(G)** a plane in the [1$\bar{1}$0]direction, with two Dirac cones on the (110) mirror invariant line. Band structures at zero flux of **(H)** a (100) wire with a size of 46 atoms in height and width, and of **(I)** a (110) wire with a size of 28 atoms in height and width. The spectra are symmetric around $k_z = \pi$, with all bands doubly degenerate.

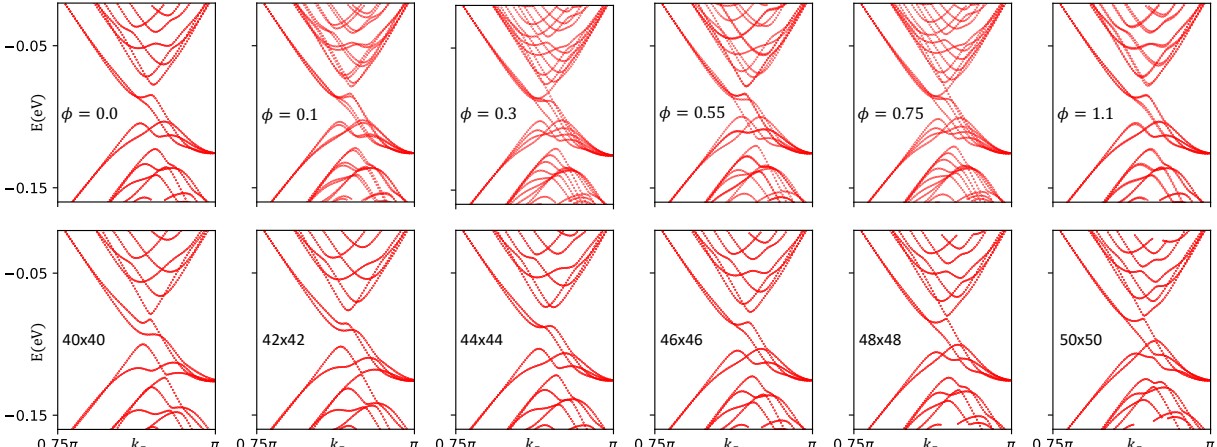

**Fig. 2. Spectrum of (100) wire.** Zoom in on low energy bands in the (100) wire's spectra. **First row:** Gap closing and re-opening under flux, of a 46x46 atoms wire. At zero flux, there are gapes between the surface and hinge bands (although the hinge states are extended to 2D close to the gap, see text). When turning on the flux, the degeneracy is removed. At certain flux values, around $\phi = 0.3$ and $\phi = 0.7$ flux quantum, a gap closing is observed between the surface and hinge bands. Around half flux quantum, most but not all bands restore their double degeneracy. Around one flux quantum, the spectrum without flux is restored. These values are close to, and not precisely at half and one flux quantum due to finite size effects, see text. **Second row:** Spectra of square (100) wires with varying sizes, at zero flux. The numbers on each figure represent the number of atoms in height and width of each wire. The dimensions of the wire affect the gaps between the surface and hinge bands.

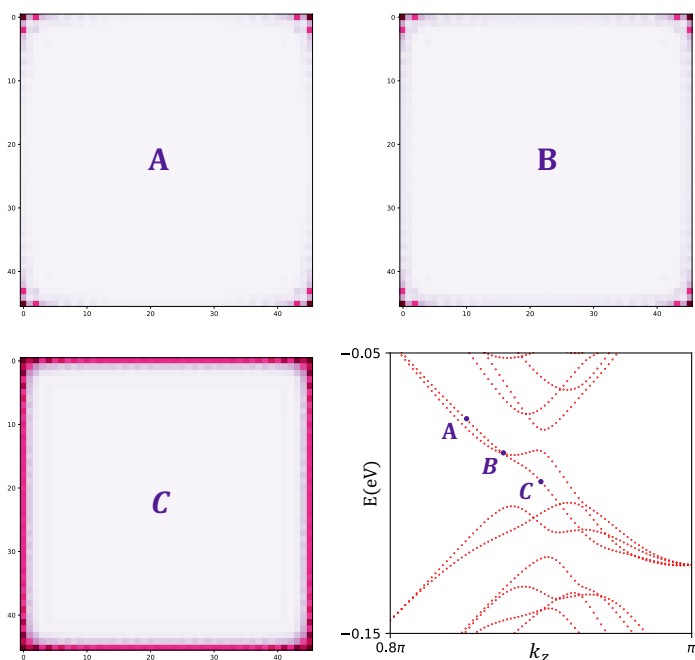

**Fig. 3. Hinge wavefunctions of (100) wire.** Spread of wavefunctions of the hinge bands on the cross section of the 46x46 (100) wire, marked A-C. As can be seen, away from the surface gap, the wavefunctions are localized in the corners of the wire. Close the gapped Dirac cone, the wavefunction is spread across the two-dimensional surfaces of the wire.

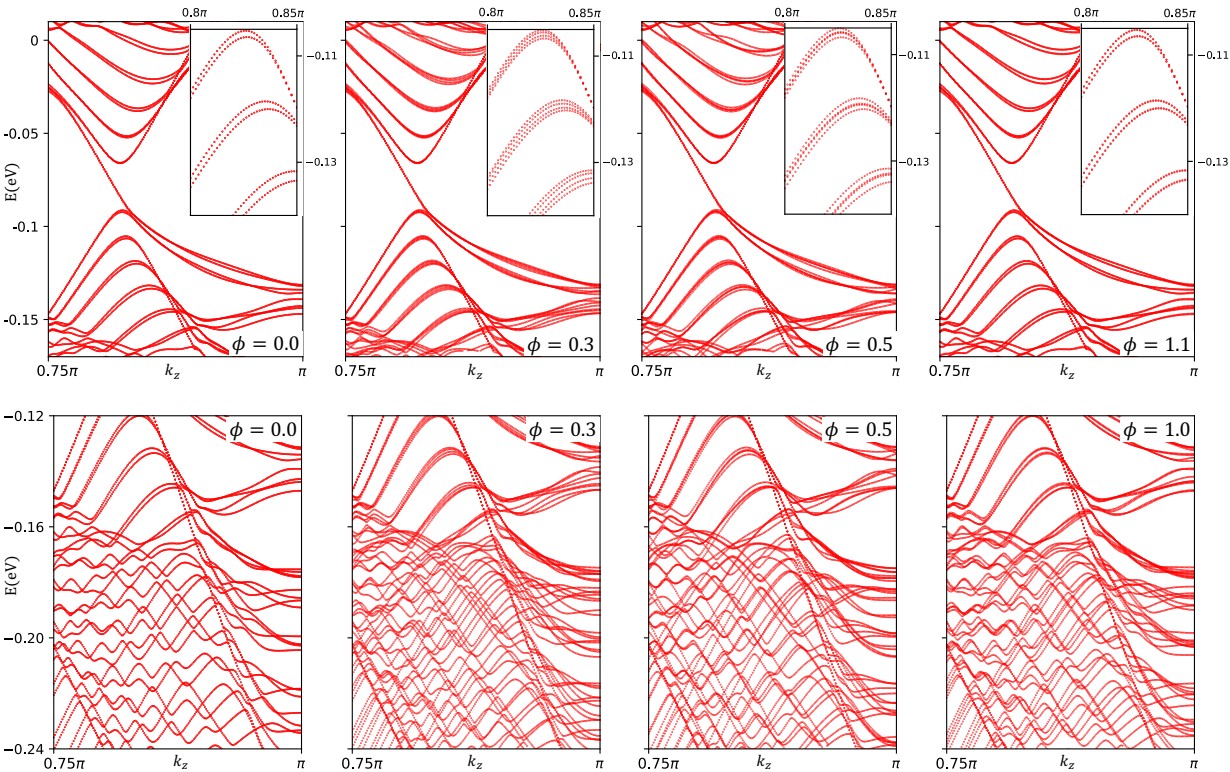

**Fig. 4. Spectrum of (110) wire with flux.** Band structure of the (110) wire with flux. In all figures, the wire is square with 28 atoms in the outermost layer in both dimensions. Flux value is indicated on each figure. **First row:** Low energy bands, showing a very weak response the flux. **Insets**: Zoom-in on the lower bands reveal groups of four bands that change degenerate partners under flux sweep. **Second row:** High energy bands, showing a strong response to flux.

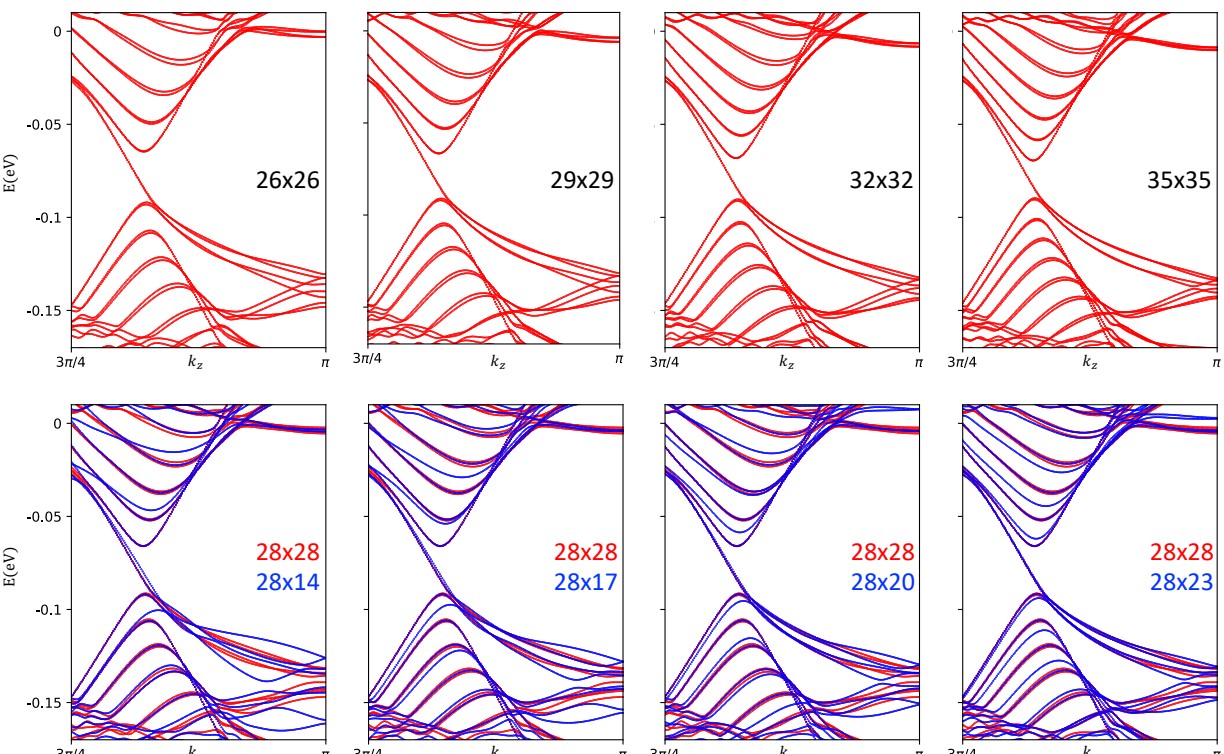

**Fig. 5. Spectrum of** $(\mathbf{110})$ **wire with varying sizes. First row:** Spectra of square wires with various sizes, the numbers indicate the number of atoms in the outermost layers in both finite directions. The spectra are similar in features, with the bands get denser for increasing sizes. **Second row:** Spectra of rectangular wires with a fixed width of 28 atoms and increasing height (blue), compared to the spectrum of a square wire with 28 atoms in width and height (red). Some of the bands do not change with size, and are located close to the four nearly degenerate bands of the 28x28 wire bands, these bands originate from the two facets of width 28 atoms. Other bands get denser as the size of the rectangular wire increases, they originate from the facets with increasing width. When two doubly degenerate bands overlap, a small mixing occurs.

# Supplementary Materials for

## Confined vs. extended Dirac surface states in topological crystalline insulator nanowires

Roni Majlin Skiff, Fernando de Juan, Raquel Queiroz, Subramanian Mathimalar, Haim Beidenkopf*, Roni Ilan*

*Corresponding authors. Email: haim.beidenkopf@weizmann.ac.il, ronilan@tauex.tau.ac.il

**This PDF file includes:**

Supplementary Text
Figs. S1 to S2

**Supplementary Text**

``Toy model" of four sites in a circle

To support this picture of slightly mixed copies of the same surface theory we first show that band mixing and pair switching can be captured by a simple ``toy model" of four sites, each site representing a single surface with an identical on-site energy $\varepsilon$ and a coupling that gives rise to hopping terms between the sites, $t$, in a circle. To this system we also add a Berry phase of $\pi$, incorporating the effect of spin momentum locking which is reflected in such a phase that is acquired when completing a closed loop around the wire. Such a system is described by the simple Hamiltonian

$$H = \begin{pmatrix} \varepsilon & -t & 0 & -te^{i\pi} \\ -t & \varepsilon & -t & 0 \\ 0 & -t & \varepsilon & -t \\ -te^{-i\pi} & 0 & -t & \varepsilon \end{pmatrix} \tag{3}$$

Solving for the energies of the system, we observe that the fourfold degenerate on-site energy is split to two twofold degenerate energy levels: $E_{1/2} = \varepsilon - \sqrt{2}t, E_{3/4} = \varepsilon + \sqrt{2}t$. Adding a flux as an additional phase to the hopping parameters removes the degeneracy. At half flux quantum (an additional $\pi$ phase compensates for the Berry phase) there are three energy levels, with the middle one doubly degenerate: $E_1 = \varepsilon - 2t, E_{2/3} = \varepsilon, E_4 = \varepsilon + 2t$. This is precisely the behavior a sub-space of four bands in the surface spectrum of the (110) experiences under flux tuning. We therefore conclude that the lifted fourfold degeneracy in the spectrum results from four identical copies of the spectra of each face.

Mass configurations

The possible mass configurations at the surface of the two wires discussed in the main text must respect the symmetries of the full wires, and not just those of the surface theory. For the square (100) wire, these are additional two mirror planes that cut the wire diagonally (see Fig.S1). These mirror planes are those protecting hinge model in the HOTI phase (5). In the (110) wire, two mirror planes cross the middle of the two facets (see Fig.S1). In addition, in square or rectangular geometries there is a $C_4$ and $C_2$ discrete rotation symmetry with respect to the wire's axis.

For the purpose of the discussion here, we consider the quasi-one-dimensional low energy band structure of the surface bands with $k > 0$, where $k$ is the momentum along the wire's axis. It is sufficient to consider positive $k$ only since we discuss the time-reversal invariant system.

The (110) wire has a single Dirac cone. We now consider what mass configurations are possible following symmetry constraints. The mass term can break time-reversal as shown in the main text, since the cone at negative momentum will have a similar mass but opposite in sign. Considering a square wire, there is one mass configuration that respects both $C_4$ and mirror symmetry, which is that of masses of alternating signs as depicted in the Fig.S1. In this configuration, masses have equal magnitude but opposite signs when approaching the corners on a single facet of the wire, and are also constrained to switch sign when traversing a corner.

In the (100) square wire there are two Dirac cones at $k > 0$, for which the mass profile is opposite in sign as constrained by mirror symmetry. The combination of $C_4$ and the diagonal mirror symmetries enforces that per Dirac cones, the mass at each facet has either a positive or a

negative sign, and switching sign on neighboring facets (see Fig.S1). This is because $C_4$ interchanges the two Dirac cones.

Symmetry of spectra under flux

As mentioned in the main text, the spectra of the (100) and the (110) wires are symmetric around the point $k_z = \pi$ in the one-dimensional BZ at zero flux and remain symmetric for all flux values. For this reason, the spectral response to flux and to dimensional changes is presented in the main text for $k_z < \pi$ values only. The Spectra of a square (100) wire of size 46x46 atoms and a square (110) wire of size 28x28 atoms at various flux values around $k_z = \pi$ are presented in Fig.S2.

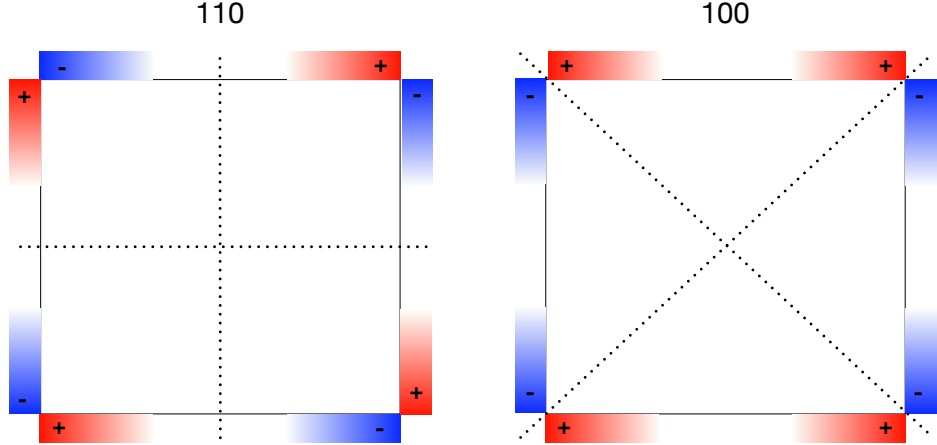

**Fig. S1.**

Two mass sign configurations respecting bulk and surface mirror symmetries as well as $C_4$ symmetry for the positive momentum surface cone of the (110) wire (left) and one of the two surface cones of the (100) wire (right). The second cone of the (100) wire replaces all blue with red, red with blue.

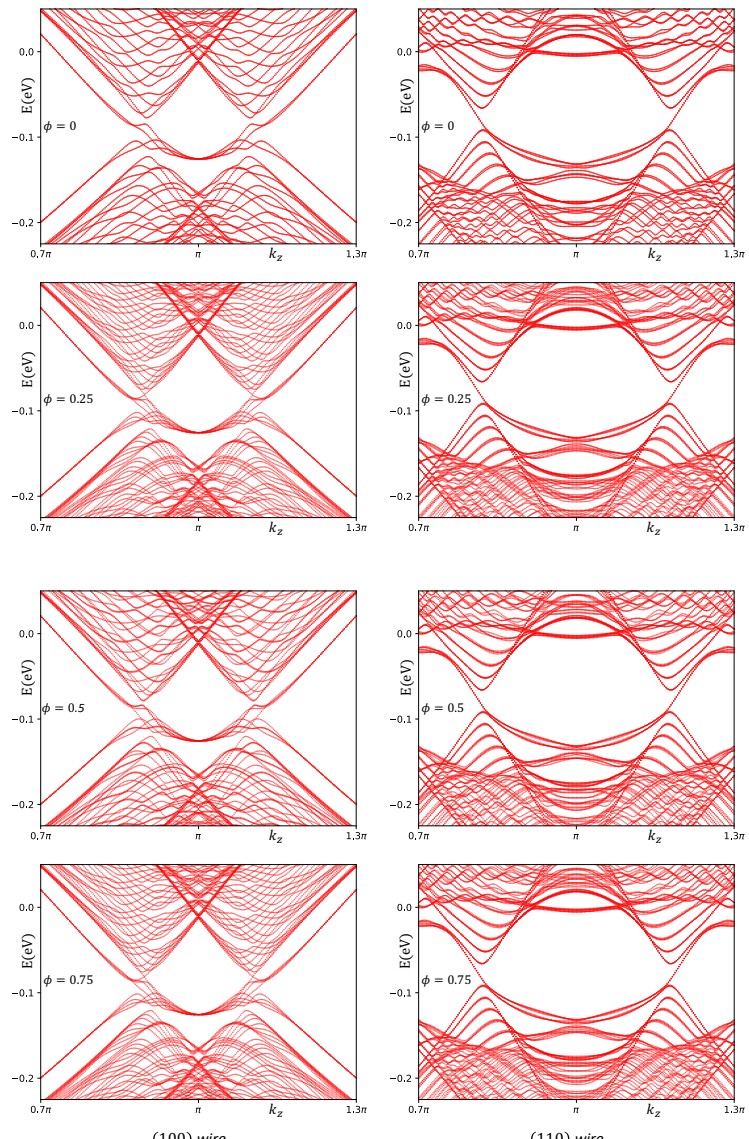

**Fig. S2.**

Spectra of the (100) wire (left column) and the (110) wire (right column), remain symmetric around $k_z = \pi$ also for non-zero flux values: $\phi = 0, 0.25, 0.5, 0.75$ for first, second, third and forth rows, respectively.