# Peer review of "Confined vs. extended Dirac surface states in topological crystalline insulator nanowires"

_SciPost Physics_

## Round 1 · Referee Report · Anonymous · 2022-4-5

Report
The authors study the flux dependence of the Dirac states in SnTe nanowires. Their new result is that wires with all four facets of the < 100 > type display Aharonov-Bohn (AB) oscillations, while in nanowires with the four facets of the < 110 > type such oscillations are absent.
First the authors utilize an oversimplified low-energy effective theory to argue that in topological crystalline insulators the AB oscillations are different from the AB oscillations in strong topological insulators. The authors correctly point out that this analysis does not take into account the important effect of the preserved/broken crystalline symmetries at the facets of the wires nor the existence of the hinges, which are very important in topological crystalline insulators. Therefore, the presented calculation is not useful for understanding the AB effect in topological crystalline insulators. Nevertheless, I still like the discussion because it helps the readers to understand the differences between topological crystalline insulators and strong topological insulators.
Next the authors study the flux response in SnTe wires. The (100) wires support both surface states and hinge states, which have been discussed in the previous works. The new effect predicted in the manuscript is the interplay of these states in the presence of magnetic flux, and the authors find that for particular (non-universal) values of magnetic flux the gap between the hinge and surface bands closes. Finally, in (110) wire the authors find that low-energy bands respond very weakly to the flux. The authors convincingly argue that in this case the low-energy bands originate from the four facets so that there exists only weak effective couplings between them.
The results are obtained using standard straightforward calculations and the effects originate from the known surface and hinge states of SnTe nanowires. All the results are consistent with the existing literature. Thus, I do not see any reason doubt the correctness of the results and the manuscript can be considered for publication in some form.
However, in order that the manuscript can be accepted for publication in Scipost Physics it should also satisfy one of the expectations:
1) Detail a groundbreaking theoretical/experimental/computational discovery;
2) Present a breakthrough on a previously-identified and long-standing research stumbling block;
3) Open a new pathway in an existing or a new research direction, with clear potential for multipronged follow-up work;
4) Provide a novel and synergetic link between different research areas.
The results are not theoretically surprising given that there already exists a very good understanding of the surface states and hinge states in SnTe wires based on the previous theoretical works. Moreover, the new results are quite specific to SnTe materials. Therefore, I do not consider the expectations 1), 2) and 4) relevant in this context. On the other hand, the manuscript could potentially satisfy the requirement number 3) if the authors could convincingly argue that their results are relevant for the experiments trying to probe the interplay of surface states and hinge states in SnTe nanowires. Unfortunately, the current manuscript falls short in providing actual predictions for the experimental observables. In order that I could recommend the manuscript for publication in SciPost Physics, the authors need to complement their work with at least two additional calculations.
1) Currently, the authors only discuss the AB effect on the level of the spectrum but do not discuss how it could be experimentally observed. To satisfy the expectations of SciPost Physics, the authors should compare the magnetic field-dependence of the conductance in the cases of low-energy hinge/surface bands and the high-energy bands.
2) In the current version of the manuscript the Zeeman effect is neglected. Some experimental works have reported g-factor on the order g~50 in SnTe materials. Therefore, it seems likely that the Zeeman effect would be relevant in the experiments and it should be considered in the manuscript.
Author: Roni Majlin Skiff on 2022-09-22 [id 2840]
(in reply to Report 1 on 2022-04-05)
We thank the referee for reviewing and commenting on our work.
We agree that in hindsight our results make perfect sense and are consistent with the existing literature. However, we would like to point out that one of our key results, namely that of Dirac surface state confinement due to geometry and symmetry considerations in wires, was not previously known. The presence or absence of AB oscillations is our way to diagnose such an effect, but of course, the consequences are greater than the fate of AB oscillations and are relevant for surface state design patterns. We have expanded on this in the introduction of the paper. In this we stress that although our predictions for AB oscillations are important and can in principle be measured, confinement can have other consequences as well.
Regarding the additional information, we address the points raised below:
“1) Currently, the authors only discuss the AB effect on the level of the spectrum but do not discuss how it could be experimentally observed. To satisfy the expectations of SciPost Physics, the authors should compare the magnetic field-dependence of the conductance in the cases of low-energy hinge/surface bands and the high-energy bands.”
As stated above, the main goal of the paper is to present the mechanism of confinement of Dirac fermions due to spatial symmetry breaking in TCIs, using a comparison between two representative cases of confined vs. unconfined surface states. The data are presented in order to support this claim. While a calculation of conductance as a function of magnetic field could be useful for future experimental works, we believe it is beyond the scope of this work.
In addition, we stress that there already exists literature on AB oscillations in strong TI wires, both experimental and theoretical, so an exploration of transport in TCI may not add surprising results. It is, however, technically non trivial since symmetry considerations in this paper would require an extensive discussion if one wants to represent disorder effects, as we would like to do for the sake of completeness. Such a discussion would overburden the current manuscript and take away from our main message so we believe it is better left for a followup work.
With that said, the predictions of AB responses in the different wire setups can be ideally and directly probed in scanning tunneling microscopy of SnTe nanowires. Using spectroscopic mapping both the evolution of induced surface gaps and the formation of hinge states, as well as the interplay among them, can be visualized as a function of threaded magnetic field. Our theoretical results thus mark a clear and unique path for the exploration and characterization of topological boundary modes in experiment. This is now conveyed in the Summary and Conclusions section of the paper.
“2) In the current version of the manuscript the Zeeman effect is neglected. Some experimental works have reported g-factor on the order g~50 in SnTe materials. Therefore, it seems likely that the Zeeman effect would be relevant in the experiments and it should be considered in the manuscript.”
We thank the referee for raising this point, as it surely has consequences and may affect experimental systems.
First, we stress that the magnitude of the magnetic field required to see AB oscillations in nanowires is small: considering a typical cross-section of $60\times 60 nm^2$, the field required for one flux quantum is $B=1.1 T$. This statement is independent of the type of wire, and depends only on the wire dimensions. For these field values, and for g factor g=57 measured for SnTe [https://journals.aps.org/prb/abstract/10.1103/PhysRevB.96.205129 ], the Zeeman splitting is $\frac{g\mu_{B}B}{2}=1.9 meV$, which is small compared to the sub-band spacing in our case.
Next, we point out that Zeeman coupling is also relevant for strong TI wires where it breaks TR symmetry. For example, in TI wires made from BiSbSe the g factor is estimated to be approximately 23. In experiments done on such wires, AB oscillations were observable but the details are slightly modified compared to predictions [https://www.nature.com/articles/ncomms8634#auth-Ruidan-Zhong ], as was also discussed in theoretical works (e.g https://journals.aps.org/prb/abstract/10.1103/PhysRevB.84.201105 , https://www.scipost.org/10.21468/SciPostPhys.6.5.060 ). We believe this should also be the case for our system as we now explain.
The changes induced by the Zeeman coupling will modify the details of the observed band structure, but not the main conclusions of our work. This can be seen as follows: a Zeeman term can be added to the Hamiltonian to consider how it may change the surface theory and as a result, the effects we present. We consider for example adding a Zeeman term to the low energy model of the surfaces of SnTe. Since the magnetic field is parallel to the wire, it is parallel to each one of its facets. Such a term does not gap the surface Dirac cone, but shifts their location, see ref. [https://www.nature.com/articles/ncomms1969] and specifically its correction in [https://www.nature.com/articles/ncomms2844 ]. The low energy model of SnTe surfaces as presented in this reference is of the from $\textbf{k}\times\boldsymbol{\sigma}=k_{x}\sigma_{z}-k_{z}\sigma_{x}$. If the wire is along the $z$ axis, and a Zeeman term $g\mu_BB\sigma_z$ is added, the location of the Dirac cone in the surface theory will shift along the $x$ coordinate. Performing a coordinate transformation and taking $x$ around the wire, this shift will modify the flux values needed for gap closing. In terms of the general low energy model presented in Section II of the paper, a Zeeman term will shift the location of the Dirac cones along $k_z$.
These considerations are now conveyed in section II and in the supplementary materials of the manuscript.
Author: Roni Majlin Skiff on 2022-09-22 [id 2841]
(in reply to Report 2 on 2022-04-27)We thank the referee for their encouraging remarks and are glad that the referee appreciates the novelty of our results.
We address the questions and comments raised below:
"1) One way in which this work might represent a breakthrough would be by its connection to (future) experiments. However, it is not clear what length and energy scales are required to observe such AB oscillations. For instance, [https://pubs.acs.org/doi/abs/10.1021/nl402841x] reports AB measurements in SnTe nanowires that do not appear to have a small number of well-defined facets. They claim to observe oscillations consistent with Dirac cones, which might mean that the wire is doped such that the Dirac cone states are at the Fermi level. This raises the question of how much control over the chemical potential (and over the temperature) would be required to observe the geometry dependence of the AB oscillations. Is it possible to estimate the gap due to mirror symmetry breaking, perhaps by performing a scaling analysis? If this gap is enhanced due to finite-size effects, how small should the wires be, given the available experimental temperatures? From [https://pubs.acs.org/doi/abs/10.1021/acsaelm.0c00740], it seems that diameters of around 50 nm are achievable. Is this small enough?"
In the reference mentioned, [ https://pubs.acs.org/doi/abs/10.1021/nl402841x ], the authors fabricated single-crystalline SnTe nanowires of a typical width of $59nm$, however the surfaces of the wires are not specified and therefore the surface theory is unknown. Nonetheless, AB oscillations are measured, which demonstrates that they are accessible in transport experiments, in terms of energy scales and at temperatures as high as $30K$.
Absence of AB oscillations requires fabricating a wire in a geometry that introduces mass terms in the wire’s corners, like the (110) wire presented in our paper. When the chemical potential is in the energy range that is affected by the gap induced by symmetry breaking, the bands in its vicinity will have a different response to flux. Since the gap at the corners is caused by a local perturbation due to the breaking of symmetries near the hinges, it is not expected to depend on the overall wire’s dimensions, but on the details of the hinge and the particular bulk material. An examination of the spectra of two (110) wires with different sizes shows that the energy scale in which there are no AB oscillations, namely the energy cutoff above which bands start to respond to flux in the usual way, is approximately $150meV$. In transport experiments on strong TI nanowires, gate voltage control over the chemical potential with a resolution of the order of $10meV$ was achieved. Although the level of control over the chemical potential is determined by the specific details of the experiment, such as the density of states of the bands near the Fermi level, the dimensions of the device and its fabrication process, a similar resolution in SnTe nanowires may be accessible.
Lastly, we would like to stress that transport may not be the only path to explore the band structure of the wire, and in particular, Dirac fermion confinement and corner gaps. The extended vs. confined nature of the surface states in TCI nanowires and other nanostructures can potentially also be probed by STM (e.g.https://www.nature.com/articles/nature09189 , https://www.nature.com/articles/nphys3805 ) or even ARPES (https://www.nature.com/articles/s42254-019-0088-5 https://www.nature.com/articles/srep29493 , https://journals.aps.org/prl/abstract/10.1103/PhysRevLett.110.036801).
A discussion regarding the measurement possibilities of the effects is now added to the Summary and conclusion section of the paper.
"2) I believe the accuracy of some statements might be improved. For instance, in the abstract the authors say that "wires with all four facets of the <100> type display pronounced and unique" AB oscillations. However, the magnitude of these oscillations is not quantified. Also, the claim to uniqueness seems to be at odds with the statement of page 5, where the results are described as being "compatible with predictions for AB as a sum of those arising from projections of several Dirac points with a finite size gap tunable by flux." Just after that, the authors state that "The physics is richer due to an interplay of the surface and hinge modes at the corners of the wire." Richer how, in what way? Do the hinge modes play any role for the AB oscillations?"
We thank the referee for pointing out some parts of the text that may need clarifications or modifications. Following their suggestion, some changes were made:
-We replaced the words “pronounced and unique” with the word “novel”.
-The meaning of the statement regarding predictability is that the numerical results are compatible with the predictions of the model presented in section I of the paper, not previous works. Minor changes have been made in the text to clarify that, see the details in the list of changes.
-We removed the word richer, which may indeed be confusing as we ultimately show the hinge modes merge with the 2D Dirac cones in the vicinity of the surface gap.
"3) I could not find the supplementary information files containing all the data generated and analyzed by the authors (see "Data availability" section)."
All data used for the analysis is presented in the paper itself or in the supplementary material within the figures. In addition, the code is available at https://github.com/ronimajlin/SnTeNanowire.git .
"4) I believe there may be a bug in the authors' code. In Fig. 2, it is claimed that the double degeneracy of all bands is not restored at precisely one flux quantum "due to finite size effects, see text." I could not find this discussion in the text. Further, the fact that the spectrum is not identical at ϕ=0 and ϕ=1 regardless of system size makes me suspect there is a bug in the code. Taking a Landau gauge for the vector potential would mean that the hopping, say, in the x direction (which is a 6×6 matrix connecting Sn and Te sites) would be modified as Tx→Txexp(2πiϕy), where y is the (integer) coordinate of the sites in the y direction. Working in such a gauge would mean that the Hamiltonian is identical at ϕ=0 and 1, regardless of how many sites there are in the system. The spectra should therefore be identical."
For a perfectly two dimensional surface, as is the case for an effective model that accounts for the surface theory alone, indeed the bands are expected to be identical at zero flux and at one unit of a flux quantum enclosed by the two dimensional surface. In the case of a tight binding model, the surface states have a finite penetration depth, which makes the flux values in which the bands are perfectly restored somewhat ill defined, due to making the cross-section of a finite width. This is why flux values for which certain effects occur can deviate from the predictions made based on a strictly two dimensional model. These effects were observed before in several works using tight binding models to demonstrate gap closure and the emergence of Majorana modes in strong TI wires coupled to superconductors slightly away from half flux quantum, for example. We are therefore confident that our results are correct and that this effect is not a result of a bug in our code.
This explanation appears in the text, in the sentence before the last paragraph of Section .III.A - “The reason this degeneracy is restored close to, and not exactly at half flux quantum, is due to the finite penetration depth of the surface states into the bulk.” - along with citations to papers which show and explain this point as well.
Note that while an effective Hamiltonian of the surface at one flux quantum will be identical to that at zero flux, in a 3D tight binding model the bulk’s Hamiltonian will be different at zero and one flux quantum since the flux through each unit cell will be a fraction of one flux quantum.
To make this point clearer in the text, a better definition of the magnitude of the flux $\phi$ was added to Section II, as detailed in the list of changes of the new version of the manuscript.
"5) Finally, a minor point: please add references to the first paragraph."
Indeed the first paragraph is improved by adding more relevant references and we thank the referee for pointing this out.

---

## Round 1 · Referee Report · Anonymous · 2022-4-27

Strengths
1 - The discussion is well-structured, with a simple low energy calculation helping the reader make sense of the later, tight-binding results.
2 - The symmetry analysis helps to justify the numerically obtained results, thus strengthening their validity
Weaknesses
1 - The connection to experiment is not made sufficiently clear
2 - There is some missing information in the paper, and I suspect there might be a bug in the tight-binding code
Report
The authors study the surface states of SnTe nanowires using a simplified tight-binding model, focusing on the effect of an orbital magnetic field, which is expected to lead to different Aharanov-Bohm oscillations depending on the nanowire geometry. The authors begin their discussion with a low-energy toy model calculation, which serves to give intuition on the tight-binding calculations that follow. Afterwards, a symmetry analysis is used to further justify the numerical results.
The results are novel, valid (with the possible exception of point 4 below), and clearly presented. In my opinion, however, this work does not fulfill the acceptance criteria of SciPost Physics (see https://scipost.org/SciPostPhys/about, and my first comment below). My current recommendation is to publish in SciPost Physics Core.
I have a few questions and comments:
1) One way in which this work might represent a breakthrough would be by its connection to (future) experiments. However, it is not clear what length and energy scales are required to observe such AB oscillations. For instance,
[https://pubs.acs.org/doi/abs/10.1021/nl402841x] reports AB measurements in SnTe nanowires that do not appear to have a small number of well-defined facets. They claim to observe oscillations consistent with Dirac cones, which might mean that the wire is doped such that the Dirac cone states are at the Fermi level. This raises the question of how much control over the chemical potential (and over the temperature) would be required to observe the geometry dependence of the AB oscillations. Is it possible to estimate the gap due to mirror symmetry breaking, perhaps by performing a scaling analysis? If this gap is enhanced due to finite-size effects, how small should the wires be, given the available experimental temperatures? From [https://pubs.acs.org/doi/abs/10.1021/acsaelm.0c00740], it seems that diameters of around 50 nm are achievable. Is this small enough?
2) I believe the accuracy of some statements might be improved. For instance, in the abstract the authors say that "wires with all four facets of the <100> type display pronounced and unique" AB oscillations. However, the magnitude of these oscillations is not quantified. Also, the claim to uniqueness seems to be at odds with the statement of page 5, where the results are described as being "compatible with predictions for AB as a sum of those arising from projections of several Dirac points with a finite size gap tunable by flux." Just after that, the authors state that "The physics is richer due to an interplay of the surface and hinge modes at the corners of the wire." Richer how, in what way? Do the hinge modes play any role for the AB oscillations?
3) I could not find the supplementary information files containing all the data generated and analyzed by the authors (see "Data availability" section).
4) I believe there may be a bug in the authors' code. In Fig. 2, it is claimed that the double degeneracy of all bands is not restored at precisely one flux quantum "due to finite size effects, see text." I could not find this discussion in the text. Further, the fact that the spectrum is not identical at $\phi=0$ and $\phi=1$ regardless of system size makes me suspect there is a bug in the code. Taking a Landau gauge for the vector potential would mean that the hopping, say, in the $x$ direction (which is a $6\times 6$ matrix connecting Sn and Te sites) would be modified as $T_x \to T_x \exp( 2\pi i \phi y)$, where $y$ is the (integer) coordinate of the sites in the $y$ direction. Working in such a gauge would mean that the Hamiltonian is identical at $\phi=0$ and $1$, regardless of how many sites there are in the system. The spectra should therefore be identical.
5) Finally, a minor point: please add references to the first paragraph.
Requested changes
See report above.

---

## Editorial Decision

unknown